# FAST.Farm load validation for single wake situations at alpha ventus

Matthias Kretschmer[1], Jason Jonkman[2], Vasilis Pettas[1], and Po Wen Cheng[1]

[1]Stuttgart Wind Energy (SWE) at Institute of Aircraft Design, University of Stuttgart, Allmandring 5b, 70569 Stuttgart, Germany

[2]National Renewable Energy Laboratory, 15013 Denver West Parkway, Golden, CO 80401, USA

**Correspondence:** Matthias Kretschmer (kretschmer@ifb.uni-stuttgart.de)

**Abstract.** The main objective of the presented work is the validation of the simulation tool FAST.Farm for the calculation of power and structural loads in single wake situations; the basis for the validation is the measurement data base of the operating offshore wind farm alpha ventus. The approach is described in detail and covers calibration of the aeroelastic turbine model, transfer of environmental conditions to simulations, and comparison between simulations and adequately filtered measurements. It is shown that FAST.Farm accurately predicts power and structural load distributions over wind direction with discrepancies of less than $10\%$ for most of the cases compared to the measurements. Additionally, the frequency response of the structure is investigated and it is calculated by FAST.Farm in good agreement with the measurements. In general, the calculation of fatigue loads is improved with a wake-added turbulence model added to FAST.Farm in the course of this study.

## 1 Introduction

Wind conditions inside a wind farm are strongly influenced by the interaction of individual turbines with the atmospheric boundary layer. In particular, wake effects of upstream located turbines affect the inflow conditions of downstream turbines. FAST.Farm is a new numerical tool developed by the National Renewable Energy Laboratory (NREL), which simulates wake effects and predicts power output as well as structural loads of turbines within wind farms. It implements the dynamic wake meandering (DWM) model originally described by Larsen et al. (2008), but with further model advancements.

The original DWM model and modified versions of it were verified and validated in previous studies. Larsen et al. (2013) published results of a validation with respect to loads and power production for single and multiple wake scenarios. In an additional validation study, Larsen et al. (2017) took a closer look at tower loads and showed dependencies of wake loads on turbine spacing. Improvements to the DWM model were suggested by Keck et al. (2014) with respect to atmospheric stability and Keck et al. (2015) regarding atmospheric shear and turbulence build-up in a wind farm. In both studies, they validated their improvements by comparing them to power production measurements as well as results of large eddy simulations (LES) in terms of velocity deficit and turbulence intensity (TI) profiles. Recently, validation efforts of the DWM model were conducted by incorporating flow field measurements from a lidar device (Conti et al. (2021)). Reinwardt et al. (2020) derived new calibration factors by using lidar measurements and validated their results in terms of power production and structural loads (Reinwardt et al. (2021)).

FAST.Farm was validated with LES in the prediction of wake characteristics, turbine power, and structural loads for a three turbine case by Shaler and Jonkman (2020). They showed that FAST.Farm calculates results in good agreement with the reference LES for most analyzed quantities. However, in low ambient turbulence conditions, higher differences were observed, which were attributed to a missing wake-added turbulence feature in FAST.Farm in this study. Shaler et al. (2020) performed a validation of FAST.Farm against full-scale data of the turbine's supervisory control and data acquisition (SCADA) system
(generator power, rotor speed, blade pitch) of a five turbine configuration. Despite problems with the used generic controller, FAST.Farm captured the trends of the measurements with good accuracy. In the Scaled Wind Farm Technology (SWiFT) benchmark study by Doubrawa et al. (2020), FAST.Farm calculated flow characteristics of a single wake in good agreement with the other considered simulation tools. Underperformance in capturing wakes compared to flow measurements were predominantly traced to inaccuracies in the inflow modeling.

In the present study, we compare results from FAST.Farm against full-scale measurement data from the offshore wind farm alpha ventus. The overall objectives are summarized as: 1) validating FAST.Farm for power and structural load predictions in single wake situations, 2) providing a detailed path on how to perform load validation with field data of an operational wind farm, and 3) providing insights into load characteristics of an offshore wind turbine subjected to single wake conditions. A one-to-one approach is followed for the validation concept. Here, environmental conditions measured at the meteorological
mast FINO1 are aggregated and directly fed as inputs into the simulations; this means that each measured 10-min event is represented in the simulations with its unique environmental parameter combination.

## 2    Methods and data

### 2.1    FAST.Farm overview

FAST.Farm is a multiphysics engineering software tool that accounts for wake interaction effects on turbine performance and
structural loading within wind farms. FAST.Farm is an extension of the NREL software OpenFAST, which solves the aero-hydro-servo-elasto dynamics of individual turbines. FAST.Farm extends this analysis to include wake effects in wind farms. As in OpenFAST, rotor aerodynamics in FAST.Farm are modeled using the blade-element-momentum (BEM) theory with options for advanced corrections, such as the inclusion of unsteady aerodynamics. Wake aerodynamics in FAST.Farm are based on the DWM model, Larsen et al. (2008), but expands on it to address many limitations of past DWM implementations.
Using the DWM method, the wake deficit behind each turbine in the wind farm is computed quasi-steadily, with the wake evolution solved via the thin shear-layer approximation of the Navier-Stokes equations in axisymmetric coordinates. Turbulence closure is captured through an eddy-viscosity model, including the influence of ambient turbulence and the wake shear layer. The wake deficit at the rotor is based on the low-pass time-filtered and azimuth-averaged radially dependent thrust coefficient calculated by OpenFAST. Wake-expansion in the pressure-gradient zone is solved with a near-wake correction. Each
wake meanders due to large-scale turbulent structures using a three-dimensional passive tracer, with a meandering velocity based on spatial averaging of the disturbed wind field. The disturbed wind field is calculated by superimposing the ambient

turbulence and the individual wake deficits from each rotor. When multiple wakes overlap, the superposition of axial wake deficits is based on a root-sum-square method.

Some of the unique innovations of FAST.Farm relative to DWM implementations in other simulation tools include:

- Improvement of wake advection, deflection, and merging;

- Calibration of wake-related model parameters against results from high-fidelity LES;

- Ability to solve all wind turbines and the farm-wide disturbed wind field in parallel;

- Optional inclusion of a wind-farm-wide super controller (not used in this paper); and

- Optional inclusion of LES-generated ambient wind data (not used in this paper).

More information on the implementation and theory behind FAST.Farm is provided by Jonkman et al. (2017), Shaler and Jonkman (2020), and Jonkman and Shaler (2021). Doubrawa et al. (2018) derived the calibration parameters of FAST.Farm by optimization with regard to LES. These constants were directly used in this study.

### 2.1.1 Implementation of wake-added turbulence

Besides modeling wake deficit and wake meandering, the DWM model by Larsen et al. (2008) includes modeling of wake-
added turbulence. This term describes the generation of turbulence behind a wind turbine rotor due to shear forces in the wake, as well as the breakdown of mainly the tip and root vortices. The contribution of wake-added turbulence to the total turbulence level inside a turbine's wake is higher for low ambient turbulence conditions (Madsen et al. (2010)). Therefore, the inclusion of wake-added turbulence is especially important for offshore conditions where ambient turbulence levels are often low (i.e. less than 10 %). Preliminary results in the course of this work (see Fig. A1) supported this observation and led to the implementation
of a wake-added turbulence feature in FAST.Farm, which was not present in previous versions. In summary, it is seen that the tower loads are strongly influenced by wake-added turbulence and the corresponding improvement of FAST.Farm leads to an increase in accuracy. In contrast, Fig. A1 indicates that the blade loads are not very sensitive to wake-added turbulence.

The herein presented implementation of wake-added turbulence in FAST.Farm follows mainly the approach by Larsen et al. (2008) and Madsen et al. (2010), which is included in the IEC 61400-1 standard (IEC (2019)). In addition to the ambient
turbulence domain, it uses a new wake-added turbulence domain defined in the meandering frame of reference. This domain is generated with Mann's spectral turbulence model (Mann (1994)), defining turbulence as homogeneous and isotropic with a length scale that equals the rotor diameter. The domain should have a fine spatial discretization to resolve the smaller turbulent scales of the wake-added turbulence.

The wake-added turbulence velocity components are scaled with the factor $k_{mt}$ defined by Eq. (1). It consists of two terms,
which are influenced by: 1) the quasi-steady wake deficit $U_{def}(x,\tilde{r}) = U(x,\tilde{r})/V_{DiskAvg}$, expressed with the wake velocity $U(x,\tilde{r})$ that is normalized by the rotor disk-averaged ambient wind speed normal to the disk $V_{DiskAvg}$, and 2) the radial wake

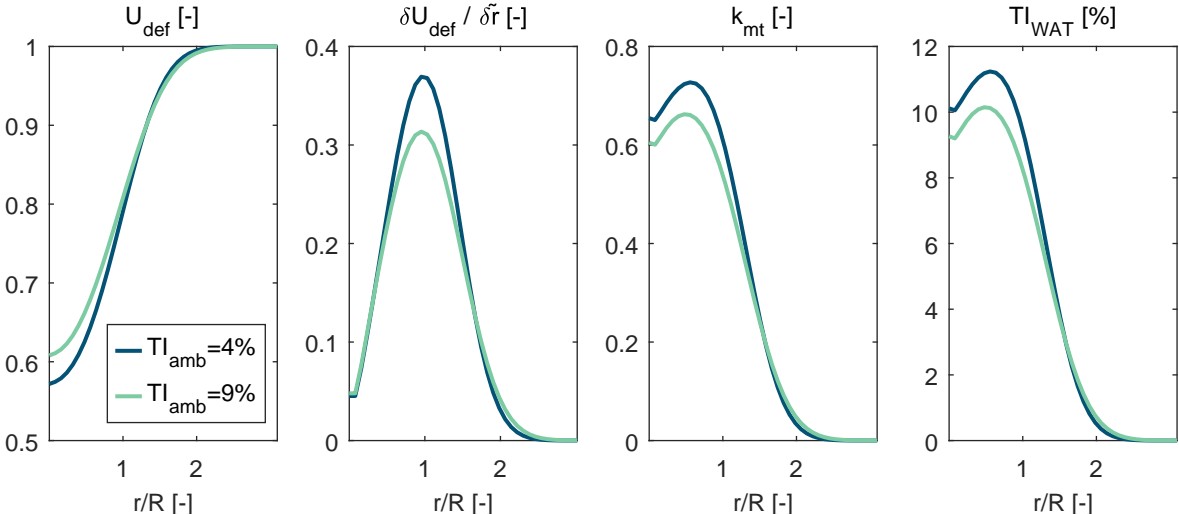

**Figure 1.** Exemplary results from the calculation of wake-added turbulence from left to right: velocity deficit distribution, radial gradient of the velocity deficit, scaling factor $k_{mt}$, resulting distribution of the turbulence intensity from the wake-added turbulence TI$_{WAT}$. Calculations are made for two ambient turbulence levels at $6\,D$ behind the rotor. Simulations are performed with FAST.Farm in neutral atmospheric conditions with a mean wind speed of $6.5\,\mathrm{m\,s^{-1}}$ at hub height.

deficit gradient. The contributions of those terms are controlled with the empirical coefficients $k_{m1}$ and $k_{m2}$. The factor $k_{mt}$ is dependent on the axial distance to the rotor $x$ and the radial location $\tilde{r}$ normalized by the rotor radius.

$$k_{mt}(x,\tilde{r}) = k_{m1} \left| 1 - \frac{U(x,\tilde{r})}{V_{DiskAvg}} \right| + \frac{k_{m2}}{V_{DiskAvg}} \left| \frac{\delta U(x,\tilde{r})}{\delta \tilde{r}} \right| \tag{1}$$

Exemplary distributions of the TI from the wake-added turbulence TI$_{WAT}$ and the corresponding velocity deficit profiles are displayed in Fig. 1. The empirical constants involved in Eq. (1) were re-calibrated to $k_{m1} = 1.44$ and $k_{m2} = 0.84$ by using the measurements. For the calibration, events with low ambient TI and close to full-wake conditions ($\pm 5°$) were utilized from the measurements as reference and damage equivalent loads (DELs) as well as the frequency response at tower base were compared. The recommended values by the IEC 61400-1 standard are $k_{m1} = 0.6$ and $k_{m2} = 0.35$. The reason for the

discrepancy between the IEC values and the re-calibrated values is not clear. However, the resulting turbulence levels from the wake-added turbulence model seen in Fig. 1 are similar to turbulence values found in the literature, e.g. from Keck et al. (2014) and Madsen et al. (2010).

The implementation of wake-added turbulence includes three major additions to the FAST.Farm code:

1. A new instance of the FAST.Farm module InflowWind is initialized for the wake-added turbulence domain; it is reused

for each turbine in the simulation domain to ensure computational efficiency. The turbulent wind box that is created beforehand with the DTU Mann turbulence generator (DTU Wind Energy (2021)) is loaded into this instance of In-

flowWind. In the course of a FAST.Farm simulation, the wake-added turbulence wind field is propagated with the ambient wind speed at hub height. The usage of a Mann turbulence field is mainly motivated by practical reasons with regard to the implementation in InflowWind; when using a Mann turbulence field, a reference wind speed can be directly defined inside InflowWind, which is eventually used to propagate the wake-added turbulence box downstream.

2. The scaling factor $k_{mt}$ is calculated in the meandering frame of reference inside FAST.Farm's "Wake Dynamics (WD)" module. The calculation is based on the quasi-steady velocity deficit and its radial gradient, that are already available inside the WD module.

3. In the module "Ambient Wind and Array Effects (AWAE)" of FAST.Farm, the velocities of the wake-added turbulence field are interpolated based on their spatial location and scaled with the spatially interpolated value of the factor $k_{mt}$. The resulting velocities are added to the ambient wind vector via vector addition in the low- and high-resolution domains of FAST.Farm and transformed from the meandering frame of reference to the fixed frame of reference.

## 2.2 Alpha ventus measurement data base

The wind farm alpha ventus is located $45\,\mathrm{km}$ north of the German island Borkum in the North Sea. It consists of twelve turbines with a rated power of 5 MW, which is shown in Fig. 2. This study focuses on the turbines AV4 and AV5, which are Senvion 5M turbines with a rotor diameter of $126\,\mathrm{m}$ and a hub height of $92\,\mathrm{m}$. They are mounted on a jacket substructure and are located approximately $6.7\,D$ apart. Within the initiative Research at alpha ventus (RAVE, 2021) measurement data from both turbines have been acquired since 2011. For example, these data were used in load validation studies for freestream conditions by Kaufer and Cheng (2013) and Popko et al. (2021). For this work, we used data from the period $01/2016 - 07/2018$ because of good availability and quality. In front of turbine AV4, the FINO1 met mast is located at a distance of approximately $3.2\,D$ providing environmental data.

### 2.2.1 Turbine measurements

The turbines are equipped with load sensors at various locations. Additionally, data from the turbine's SCADA system are available. The time resolution of all sensors is $50\,\mathrm{Hz}$. The following list explains the sensors used in this study and their calibration.

– SCADA: Generator power, generator speed and blade-pitch angle measurements are directly taken from the SCADA system.

– Nacelle yaw position: These data are also available through the SCADA system. Over longer time periods, a drift was observed in the data. This was corrected by using nacelle rotation events and correlating the known tower-base strain gauge positions with the nacelle-yaw signal. In this way, sensor offset values were derived to make the data consistent over time.

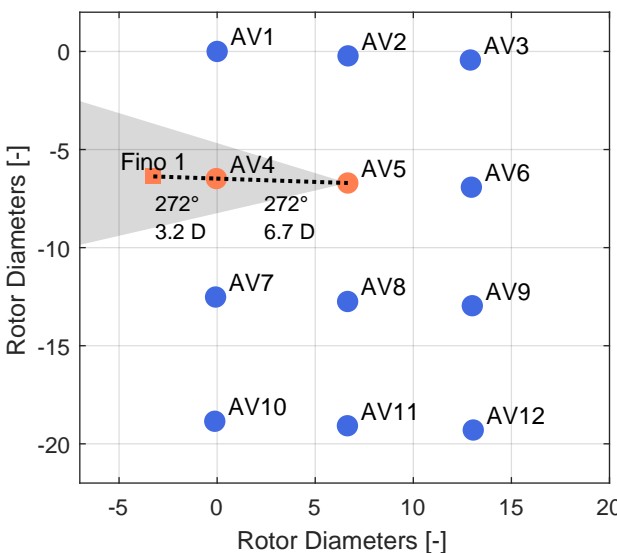

**Figure 2.** Wind farm layout of alpha ventus including FINO1 met mast. AV1-6: Type Senvion 5M. AV7-12: Type Adwen AD 5-116. North is pointing up. Shaded triangle indicates the wind direction sector $257° - 287°$ seen from AV5.

– Tower-base bending moments were calculated from strain gauges located above the transition piece. The strain gauges are placed at four locations separated by $90°$ around the tower cylinder. By combining the strain gauge measurements with the nacelle's yaw position, fore-aft (FA) and side-side (SS) bending moments were derived. Nacelle rotation events during turbine shutdown and calm wind conditions were used to determine calibration factors in terms of slope and offset.

– Blade-root bending moments in edgewise and flapwise direction are measured via four strain gauges placed near the blade root. They were calibrated with rotor idling events during calm winds as well as 10-min mean operational data. The measurements were made consistent over time by adjusting slope and offset. Strain gauge signals were combined to reduce cross-talk effects.

### 2.2.2 Environmental data

Meteorological and sea conditions are measured at the FINO1 met mast and are available as 10-min statistics. Wind speed is measured with cup anemometers at 7 locations starting at $41.5\,\mathrm{m}$ height above sea level (a.s.l.) and increasing in approx. 10-m increments to $100\,\mathrm{m}$ height a.s.l. Wind speed data are corrected for met mast shadow effects as explained by Westerhellweg et al. (2011). Vertical wind shear is described in terms of the power law exponent $\alpha$, which is derived by fitting the power law on the wind speed measurements from all available heights of the met mast.

Wind direction is taken from the wind vane located at $91.5$-m height, with the correction given by Westerhellweg et al. (2010). An additional offset of $+3\,°$ is applied on wind direction, which was derived by correlating the wake deficit of turbine AV4 with the measured wind direction at FINO1.

Atmospheric stability is estimated by using the power law shear exponent $\alpha$ and applying the limits given in Table 1. This simplified approach is motivated by Westerhellweg et al. (2014) and has the advantage of good sensor availability. It is considered to be sufficient for this study because it only serves as refinement for environmental conditions and specific analyses of atmospheric stability are excluded.

Sea state is measured in terms of significant wave height and peak wave period with a directional waverider buoy (DWR-MkIII by Datawell).

**Table 1.** Classification of atmospheric stability with power law exponent $\alpha$. The given wind field parameters $\sigma_k$ are dependent on stability and based on findings by Peña et al. (2010), which is explained in Sect. 2.4.

| Atmospheric stability | $\alpha$ | $\sigma_v$ | $\sigma_w$ |
|---|---|---|---|
| unstable | $\alpha \leq 0.07$ | $0.87\,\sigma_u$ | $0.79\,\sigma_u$ |
| neutral | $0.07 < \alpha < 0.15$ | $0.81\,\sigma_u$ | $0.68\,\sigma_u$ |
| stable | $0.15 \leq \alpha$ | $0.78\,\sigma_u$ | $0.63\,\sigma_u$ |

Around alpha ventus, new wind farms have been commissioned over the years. This changes inflow conditions at alpha ventus compared to situations where no other wind farms were in close vicinity, as shown by Pettas et al. (2021). In this work, these effects are partly taken into account because measured environmental conditions are directly transferred to simulations on a 10-min event basis.

## 2.3 Filtering approach

The measurement data are clustered in 10-min events for which the statistics are calculated to allow appropriate filtering. In particular, the following filter criteria were applied:

- Only events are considered which have a data availability of more than $99\,\%$.

- Wind direction is constrained from $257°$ to $287°$ to ensure that only wake effects of turbine AV4 affect AV5. At the boundaries of this wind direction sector, nearly freestream conditions exist for both turbines; effects from wind farm blockage are ignored.

- Both turbines are operated under normal conditions. For example, down-regulation, startup, or shutdown events are omitted.

- No yaw-action takes place during an event. Additionally, both turbines operate at similar yaw angles, allowing only events where the difference in the mean yaw position is less than $6°$.

## 2.4 Transfer of environmental conditions to simulations

An important part of a validation procedure is to ensure high quality simulation inputs. In this study, turbulent wind fields were created with the software TurbSim (Jonkman (2009)) fed by measured meteorological conditions at FINO1. The wind field generation with TurbSim applies Taylor's frozen-turbulence assumption and propagates turbulent eddies with a constant velocity while neglecting the evolution of the turbulent eddies in the downstream direction. According to Shaler et al. (2019a) who compare results from FAST.Farm with results from LES, this assumption is considered a reasonable simplification.

The von Karman turbulence spectrum is adapted with a modified wind profile: At heights where measurements at FINO1 exist, mean wind speed and TI were directly used. For heights exceeding the met mast, a power law is utilized to extrapolate mean wind speed; for TI, the standard deviation value at the top met mast position is kept constant over higher heights. It is assumed that the cup anemometer only measures turbulent fluctuations of the wind speed component u, defined as the standard deviation $\sigma_u$. The relationship of turbulent fluctuations of the three wind speed components u,v,w is then adjusted according to atmospheric stability (see Table 1). It is based on findings by Peña et al. (2010), who investigated, among other things, the anisotropy of turbulence with respect to atmospheric stability. Their values of the anisotropy parameter $\Gamma$ are translated to standard deviations by using the relationships provided by Mann (1998).

Turbulence length scale $L$ is approximated with Eq. (2) as suggested by Kelly (2018). Here, the TI measurement at $z = 90$-m height and the power law shear coefficient $\alpha$ serve as inputs.

$$L \approx z \, \frac{TI}{\alpha} \tag{2}$$

Coherence $\gamma$ of the wind speed components k=u,v,w between points $i$ and $j$ is defined according to the Davenport (1961) model in Eq. (3):

$$\gamma_{i,j} = \exp\left(-c_{\mathrm{k}} \, \frac{f \, \delta}{\bar{u}}\right), \tag{3}$$

where $f$ is the frequency, $\delta$ is the separation distance between points $i$ and $j$, and $\bar{u}$ is the mean wind speed of both points. The decay coefficient $c_{\mathrm{k}}$ is chosen according to Nybø et al. (2020) and is dependent on atmospheric stability. Note that for stable conditions, the value for coefficient $c_{\mathrm{k}}$ is used that is dedicated to neutral conditions. The coherence model applies on the wind speed components individually and ignores correlation between different velocity components (e.g. u-v correlation).

## 2.5 Calibration of aeroelastic simulation model

The aeroelastic simulation model was generated with structural and aerodynamic information provided by the manufacturer. Simulations were performed with the original turbine controller. In addition, a thorough calibration of the simulation model of both turbines AV4 and AV5 was performed to match the turbines measured load characteristic in the field as closely as possible. This involves the determination of structural damping of the first tower eigenmodes in the FA and SS direction by analyzing turbine shutdown events.

Furthermore, imbalances in the rotating system were identified by looking at the frequency response during freestream events with low TI conditions. The imbalances were introduced for each blade $i$ as variation in blade mass $B_{\mathrm{M,i}}$, variation in

blade flapwise $B_{FK,i}$ and edgewise $B_{EK,i}$ stiffness, and blade-pitch offset $B_{PO,i}$. A summary of the introduced imbalances in the simulation model is listed in Table 2. Using these values, we were able to reproduce the turbines' frequency response from the field at the desired sensor locations with satisfactory accuracy. It is noted that these imbalances do not necessarily reflect the real existing imbalances; however, this approach was regarded as the best solution overcoming missing exact blade calibration measurements such as static blade deflection tests.

**Table 2.** Identified simulation model imbalances. Values without units are factors that are multiplied with the given parameter.

|  | $B_{M,1}$ | $B_{M,2}$ | $B_{M,3}$ | $B_{FK,1}$ | $B_{FK,2}$ | $B_{FK,3}$ | $B_{EK,1}$ | $B_{EK,2}$ | $B_{EK,3}$ | $B_{PO,1}$ | $B_{PO,2}$ | $B_{PO,3}$ |
|---|---|---|---|---|---|---|---|---|---|---|---|---|
| AV4 | 1 | 0.996 | 1 | 1 | 1 | 0.95 | 1 | 1 | 0.95 | $0°$ | $-0.5°$ | $+0.5°$ |
| AV5 | 1 | 1 | 1 | 0.95 | 1 | 1 | 0.95 | 1 | 1 | $-0.5°$ | $+0.5°$ | $+0.5°$ |

## 2.6 Simulation setup and presentation of results

The information presented in the previous sections was used to set up the FAST.Farm simulations. One-to-one simulations were performed, where the measured environmental conditions of 10-min events are directly used as simulation inputs. Here, six random and uncorrelated turbulence realizations of the wind field were created. Similarly, the sea state is modeled with six random realizations of the JONSWAP spectrum using the measured significant wave height and peak wave period. The wake analysis focuses on two wind speed bins in below rated conditions: 1) wind speed $6.5 - 7.5\,\mathrm{m\,s^{-1}}$ labeled in the following with "I", 2) wind speed $9.0 - 10.0\,\mathrm{m\,s^{-1}}$ labeled with "II". These bins were chosen because they imply high rotor thrust values and hence strong wake effects. In total, 1014 FAST.Farm simulations were run for wind speed bin I and 1072 for wind speed bin II.

The numerical setup of FAST.Farm is guided by the recommendations given in Shaler et al. (2019b). The domain size in the longitudinal direction is $X = 1890\,\mathrm{m}$. In the lateral direction, the largest shift of both turbines is 220 m and occurs at the considered wind directions of $257°$ and $287°$. By considering the rotor radius and reserving additional space ($2\,D$) for the wake meandering in both directions, the domain is set to $Y = 850\,\mathrm{m}$. The domain size in the vertical direction is set to $Z = 300\,\mathrm{m}$, which includes a safety margin of $\approx 1.2\,D$ above the rotor tip for vertical wake meandering. The wakes are modeled up to a downstream distance of $8\,D$ by using wake planes with a spatial resolution of $5\,\mathrm{m}$ in the radial direction.

The TurbSim wind fields for the ambient turbulence and the low-resolution domain for resolving the wake meandering in FAST.Farm have a spatial resolution of $dy = dz = 6\,\mathrm{m}$ in the lateral and vertical directions. The spatial resolution for the wake-added turbulence domain and the high-resolution domain around the turbines is $dy = dz = 3\,\mathrm{m}$. The time step for the low-resolution domain equals $dt_{low} = 2\,\mathrm{s}$ and $dt_{high} = 0.2\,\mathrm{s}$ for the high-resolution domain, which is also used for the TurbSim wind fields for the ambient turbulence and the Mann wind fields for the wake-added turbulence. At the beginning of each simulation, a transient period of $400\,\mathrm{s}$ is removed to allow the wakes to develop and to damp initial oscillations. The TurbSim wind fields are $600\,\mathrm{s}$ long and set to be periodic in order to achieve a total simulation time of $1000\,\mathrm{s}$. The Mann wind

230    fields for the wake-added turbulence cover $3\,D$ in the longitudinal direction; turbulence is reused when points outside this box are requested.

The results in Sect. 3 and 4 are presented in terms of statistics of measured quantities, which are plotted against the wind speed and wind direction. The data are clustered in wind speed bins of the size $0.5\,\mathrm{m\,s^{-1}}$ and wind direction bins of the size $2.5°$. For each bin, the mean value plus $15^{th} - 85^{th}$ percentile range is shown. DELs are calculated according to Eq. (4):

$$\mathrm{DEL} = \left( \sum_{i=1}^{n} \frac{S_i^m}{N_{eq}} \right)^{\frac{1}{m}}, \tag{4}$$

where $S_i$ refers to the range of a load cycle and $N_{eq}$ is the number of equivalent cycles (in this work: $N_{eq} = 600$ for obtaining the 1-Hz DEL). The Wöhler exponents are defined as $m = 4$ in case of steel (tower) and $m = 10$ for composite materials (blades). The DEL is not corrected for mean load effects.

To protect proprietary data, all results are normalized by either user defined values or by values of the freestream turbine, which is indicated in the figure legend. Measurements are labeled with "AV4" and "AV5", whereas simulations are named "FFarm4" and "FFarm5" making reference to the turbine numbers 4 and 5.

## 3   Initial results

A check of the turbine characteristics during freestream conditions as well as an analysis of the environmental conditions is presented, before taking wake effects into account in Sect. 4.

### 3.1   Investigation of turbine characteristics in freestream conditions

Figure 3 illustrates results of the comparison between simulations and field measurements for close to freestream conditions. This freestream sector of $240° - 257°$ was detected beforehand by correlating loads of turbine AV5 with the wind direction. In total, 2980 simulations were conducted with the software OpenFAST, which employs the same aeroelastic model as described for FAST.Farm but without wind farm-wide effects, enabling faster simulation times. Although an individual calibration of both turbines AV4 and AV5 was conducted, the differences in the considered aggregated load quantities is negligible; hence, only the results of the AV4 calibration is shown in Fig. 3.

For the operational sensors of generator power (Fig. 3 (a)) and blade-pitch angle (Fig. 3 (b)), a discrepancy of less than $5\,\%$ is found between the data of turbines AV4, AV5 and simulations described as OpenFAST. The DEL computed for the FA bending moment at tower-base (Fig. 3 (c)) shows a close match with a difference of less than $7\,\%$ in below rated conditions $(4-12\,\mathrm{m\,s^{-1}})$. In the above rated conditions, simulations agree well with measurements from turbine AV4; the loads of turbine AV5 are up to $20\,\%$ higher compared to turbine AV4. This difference is most likely related to an imbalance in the rotor system during blade-pitch actuation. Loads at the blade root are compared by means of DEL of the bending moment in the flapwise direction in Fig. 3 (d). It is observed that measurements of both turbines as well as simulations match each other well with differences of less than $10\,\%$ for wind speeds up to $16\,\mathrm{m\,s^{-1}}$.

260      Overall, the simulations can predict the measured load quantities in freestream conditions with high accuracy. This indicates that the aeroelastic simulation model is set up appropriately and that the meteorological conditions are transferred into realistic wind fields. The exact representation of hydrodynamic excitation in terms of wave loads is considered of less importance because the substructure is quite rigid and only sensors above the sea water level are taken into account in this study.

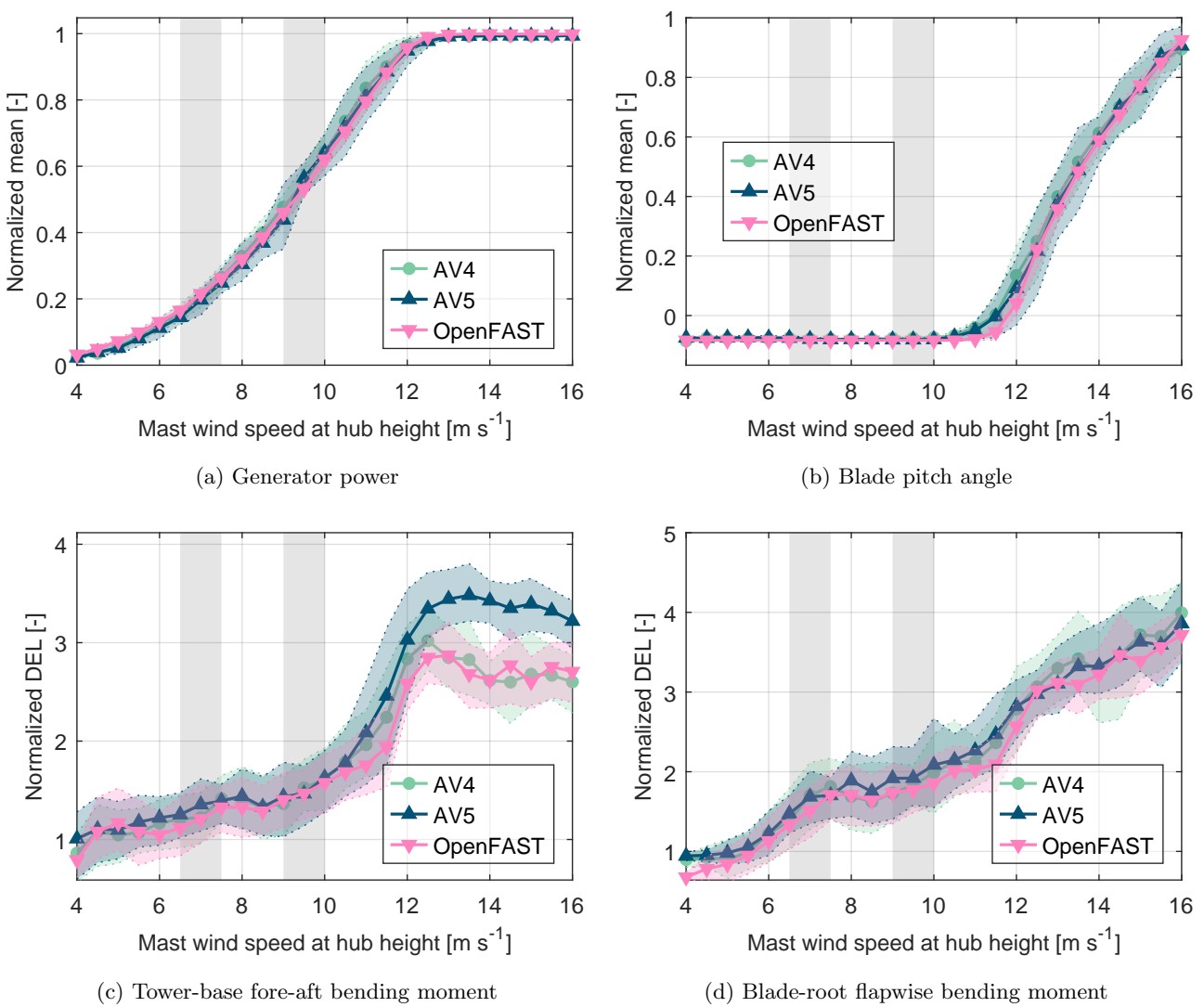

(a) Generator power

(b) Blade pitch angle

(c) Tower-base fore-aft bending moment

(d) Blade-root flapwise bending moment

**Figure 3.** Comparison of freestream (wind direction $240° - 257°$) characteristics between measurements (AV4, AV5) and simulations (Open-FAST: results are shown with calibration based on AV4). Statistics are shown as mean values per wind bin. Shaded area indicates $15^{th}$ and $85^{th}$ percentiles. Patches in grey show the wake bins I and II analyzed in Sect. 4.

## 3.2 Distribution of environmental conditions

For the analyzed wind speed bins, the TI distribution over wind direction is shown in Fig. 4. The mean TI for wind speed bin I calculates to $\approx 6.4\,\%$ and $\approx 5.9\,\%$ for wind speed bin II. Highest TI values are found in unstable atmospheric conditions whereas lowest TI values occur in stable conditions. Especially in wind speed bin II, more events are found towards wind directions from the southwest, which is the predominant wind direction. A more uniform event distribution with regard to the wind direction is found for wind speed bin I. Figure 5 depicts the distribution of the power law shear exponent $\alpha$ over wind direction. It can be seen that wind speed bin II contains higher $\alpha$ values on average compared to wind speed bin I.

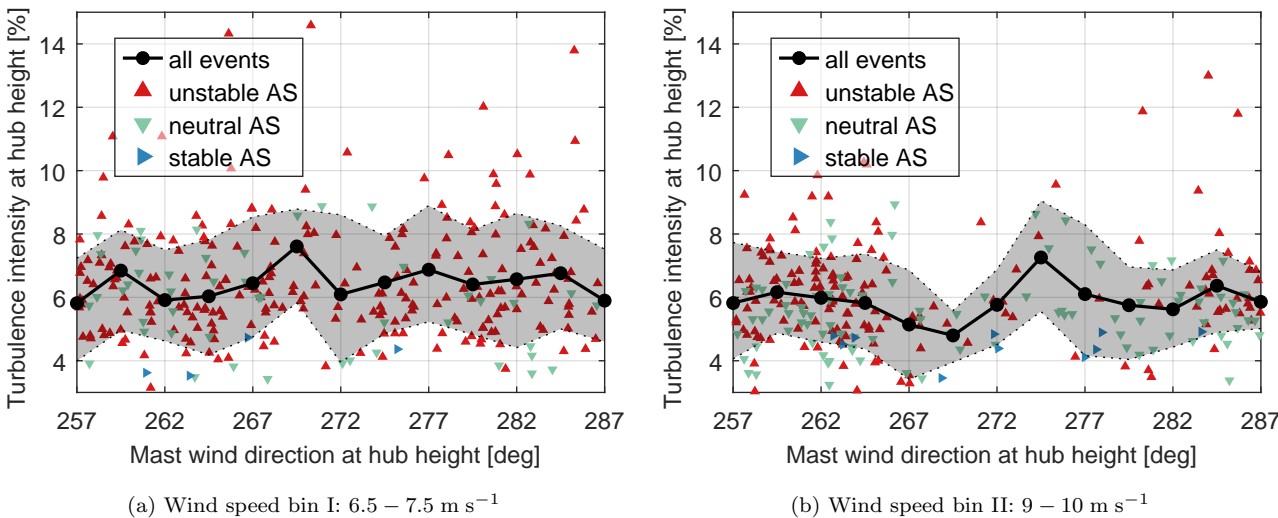

(a) Wind speed bin I: $6.5 - 7.5$ m s$^{-1}$          (b) Wind speed bin II: $9 - 10$ m s$^{-1}$

**Figure 4.** Turbulence intensity measured at 90-m height at FINO1. Statistics are shown as mean values per wind bin. Shaded area indicates $15^{th}$ and $85^{th}$ percentiles. Scatter shows TI of single 10-min events. AS = atmospheric stability.

## 4 Results of wake validation

### 4.1 Turbine performance: statistics

Results of the mean generator power prediction are presented in Fig. 6. They show a maximum power loss of $\approx 48\,\%$ in full wake conditions for wind speed bin I. At nearly $43\,\%$, power loss is slightly less for wind speed bin II due to a decreased rotor thrust coefficient compared to wind speed bin I. Wake effects are visible in the turbine power generation over a wind direction sector of $25°$. FAST.Farm is able to predict the width and depth of the power deficit with high accuracy compared to the measurements for wind speed bin I (Figures 6 (a) and (b)). For wind speed bin II, deviations of $5 - 10\,\%$ are observed in Fig. 6 (c). Relative comparisons are shown in Figures 6 (b) and (d), where results of the waked turbine are divided by the freestream results of simulations and measurements correspondingly. It is observed that relative plotting produces a better

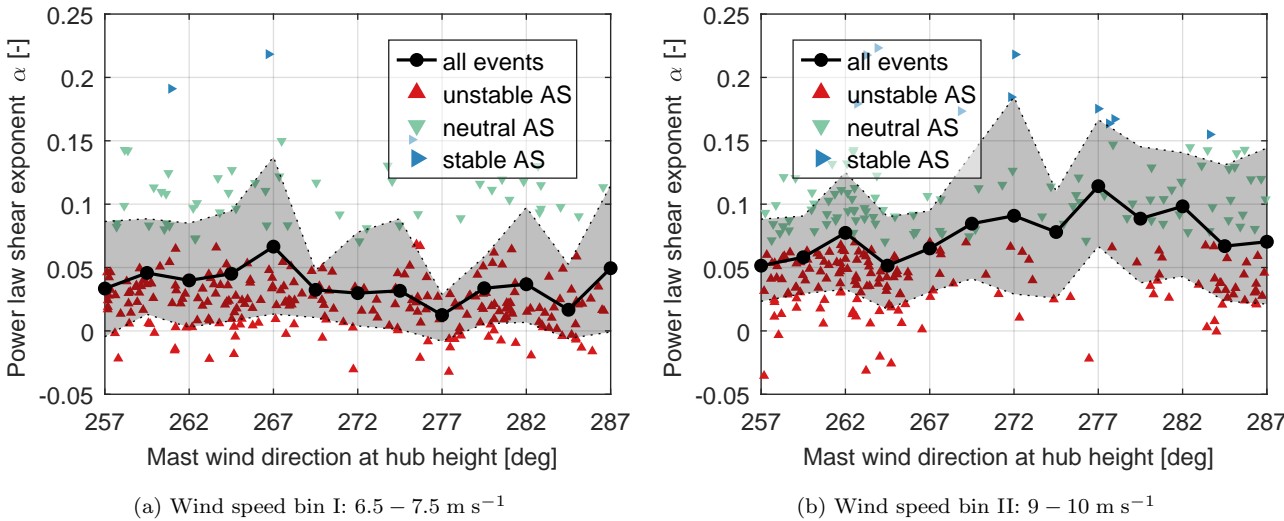

(a) Wind speed bin I: $6.5 - 7.5$ m s$^{-1}$

(b) Wind speed bin II: $9 - 10$ m s$^{-1}$

**Figure 5.** Power law shear exponent $\alpha$ derived from FINO1 measurements. Statistics are shown as mean values per wind bin. Shaded area indicates $15^{th}$ and $85^{th}$ percentiles. Scatter shows $\alpha$ of single 10-min events. AS = atmospheric stability.

match of FAST.Farm with the measurements. This is due to the decreased uncertainties potentially arising from the turbine model. Similar levels in the scatter of events indicated by the error range are found for simulations and measurements.

## 4.2 Turbine performance: detailed results

A more detailed analysis of the turbine performance is shown in Fig. 7 by plotting the probability density function (PDF) of the generator speed. The PDF was calculated for each 10-min event in the wind sector corresponding to full-wake conditions. Afterwards, mean value with error range expressed as $15^{th}$ and $85^{th}$ percentiles across the PDFs of all events were derived. In both wind speed bins, a reduction of generator speed is found for the waked turbine, caused by the wind speed deficit from the upstream turbine. In wind speed bin I, the downstream turbine operates near the cut-in wind speed, which is indicated by the peak around the normalized generator speed of 0.6 in Fig. 7 (a). From Fig. 7 (b), it can be seen that a wider range of generator speeds is covered by the waked turbine compared to the freestream turbine. This can be related to a varied operational point due to the wake deficit. Additionally, it can be partly attributed to the increased turbulence in the wake originating from wake-added turbulence as well as wake meandering. In both wind speed bins, FAST.Farm predicts the distributions from the measurements with high accuracy.

## 4.3 Structural loads: statistics

Figure 8 shows results of the fatigue loads expressed as DEL of the blade-root bending moment in the flapwise direction. By comparing the two wind speed bins, different load distributions over wind direction for the waked turbine are observed. For

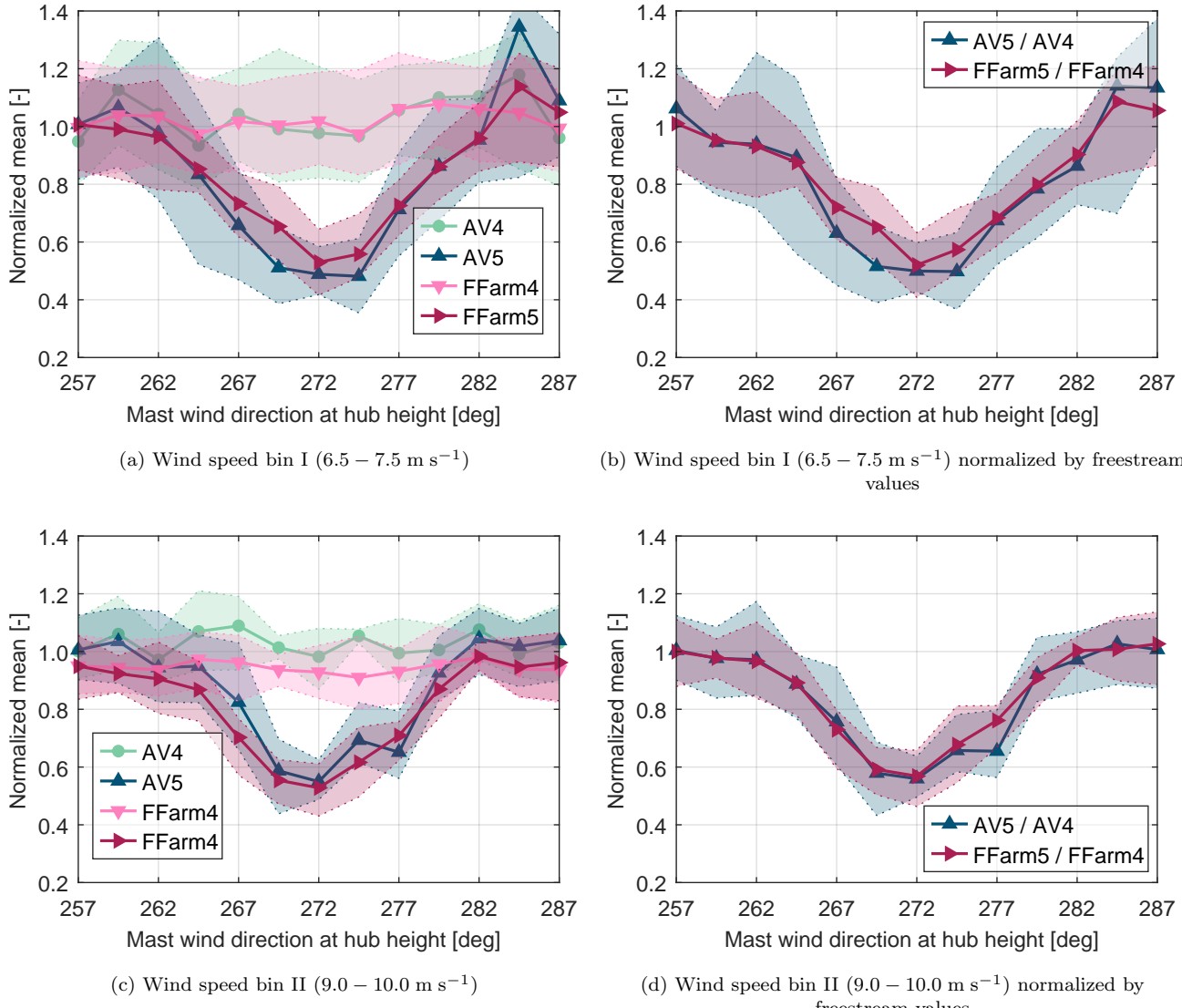

**Figure 6.** Comparison of generator power in wake conditions between measurements (AV4, AV5) and simulations (FFarm4, FFarm5). Statistics are shown as mean values per wind direction bin. Shaded area indicates $15^{th}$ and $85^{th}$ percentiles.

wind speed bin I (Fig. 8 (a)), a dip in the DEL for the downstream turbine occurs around full wake conditions at $272°$); this is not visible for wind speed bin II (Fig. 8 (b)). Influencing factors on the load distribution for the waked turbine are the wind direction and connected mean wake position, magnitude of wake meandering, ambient wind conditions, and operational point of the turbine. Varying combinations of these effects lead to different load distributions. With the chosen one-to-one simulation approach, the aim was to reduce the uncertainty arising from the different combinations. Hence, it is seen for both wind speed


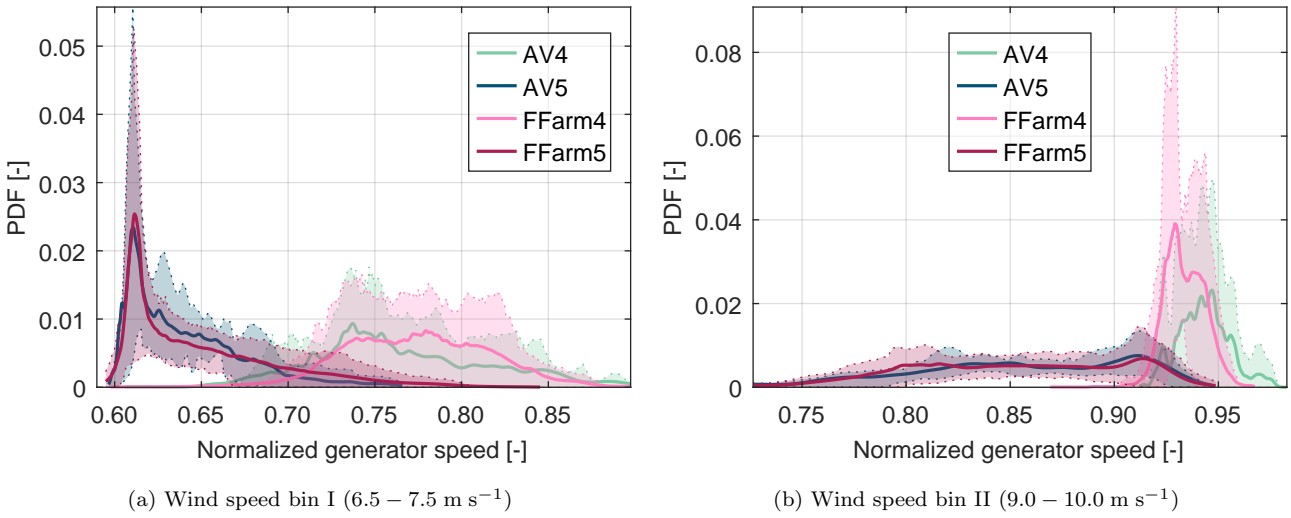

(a) Wind speed bin I ($6.5 - 7.5 \text{ m s}^{-1}$)  (b) Wind speed bin II ($9.0 - 10.0 \text{ m s}^{-1}$)

**Figure 7.** Probability density function (PDF) of generator speed for measurements (AV4, AV5) and simulations (FFarm4, FFarm5) in the wind direction sector $270.75° - 273.25°$. Thick lines show the mean of the PDFs of all considered events. Shaded area indicates $15^{th}$ and $85^{th}$ percentiles.

bins that FAST.Farm agrees well with the measurements and predicts the increase in loads and trends over wind direction with good accuracy. Overall, a load increase of approximately factor $1.7$ (wind speed bin I) and factor $2.0$ (wind speed bin II) are identified for the waked turbine.

Figure 9 displays the DELs of the tower-base bending moment in the FA direction. In contrary to the fatigue loads at the blade-root, a higher increase in loads for the waked turbine is observed for wind speed bin I compared to wind speed bin II. In particular, DELs are increased by factor $\approx 2.4$ (wind speed bin I) and factor $\approx 2.0$ (wind speed bin II) for the waked turbine compared to the DELs of the freestream turbine. FAST.Farm produces results in good agreement with the measurements in terms of magnitude and wind direction dependency. For most of the considered wind directions, the discrepancy in the mean value per bin between FAST.Farm and the measurements is less than $10\%$; in some wind directions, the difference is increased to $\approx 25\%$. The uncertainty range per bin indicated by the percentile range is predicted by FAST.Farm with good agreement to the measurements.

## 4.4 Structural loads: frequency response

Figure 10 depicts the frequency response of the structure at the tower base and blade root. Only events during nearly full wake conditions (wind direction $270.75° - 273.25°$) in wind speed bin I were considered. For each 10-min event, the power spectral density (PSD) was calculated. Then mean values for each frequency with corresponding uncertainty range expressed as $15^{th}$ and $85^{th}$ percentiles were determined. They are shown for the blade-root bending moment in the flapwise direction (Fig.10 (a)) and the tower-base bending moment in the FA direction (Fig. 10 (b)).

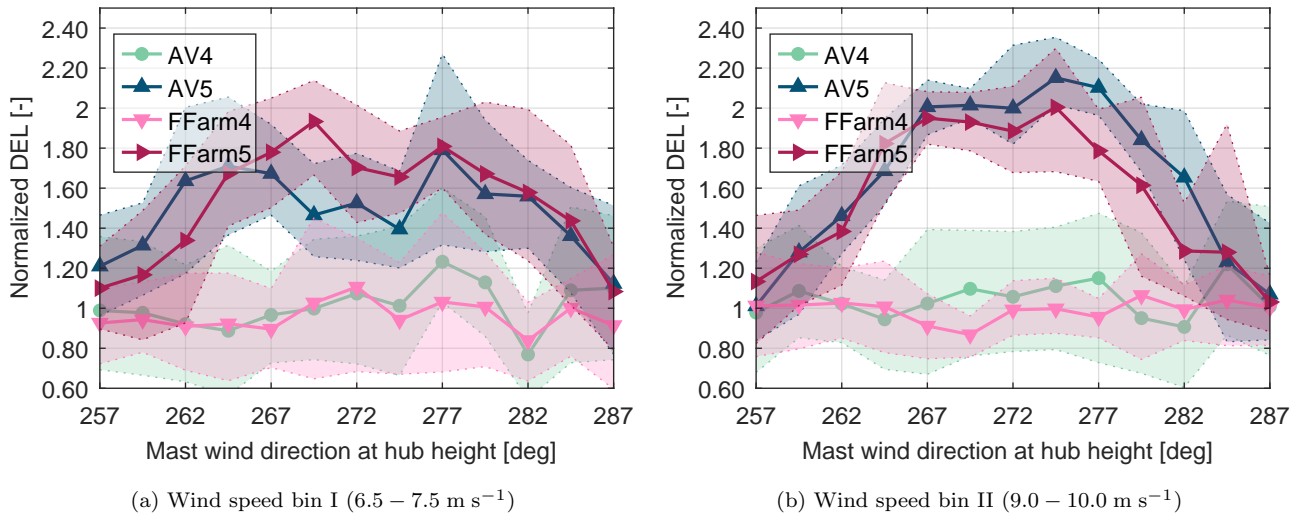

**Figure 8.** Comparison of the blade-root bending moment in the flapwise direction in waked conditions between measurements (AV4, AV5) and simulations (FFarm4, FFarm5). Statistics are shown as mean values per wind direction bin. Shaded area indicates $15^{th}$ and $85^{th}$ percentiles.

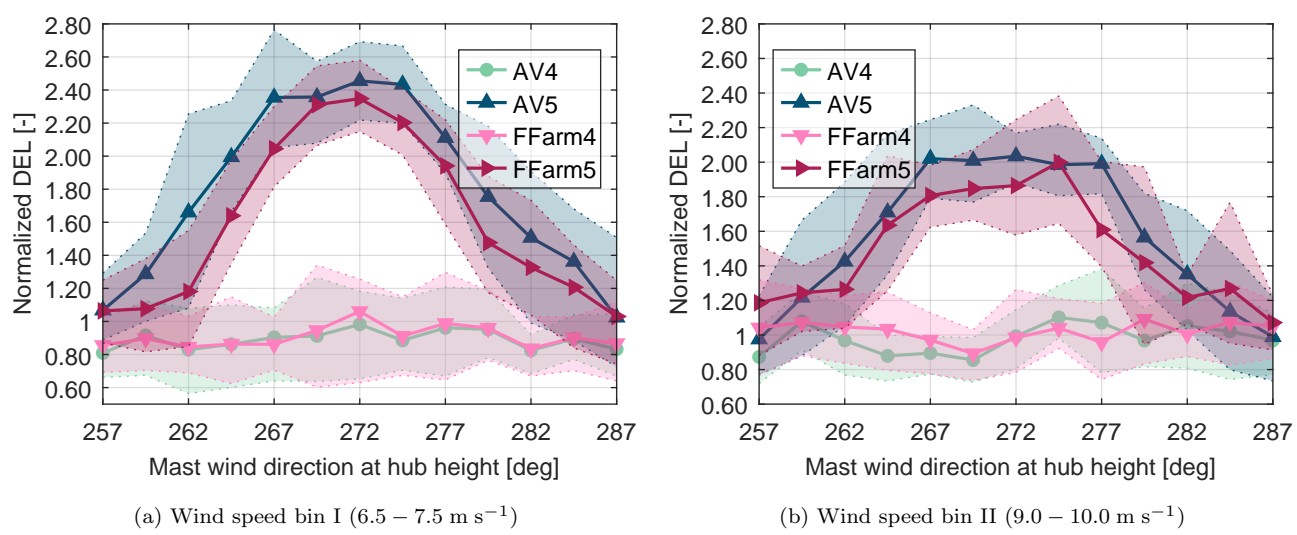

**Figure 9.** Comparison of the tower-base bending moment in the fore-aft direction in waked conditions between measurements (AV4, AV5) and simulations (FFarm4, FFarm5). Statistics are shown as mean values per wind direction bin. Shaded area indicates $15^{th}$ and $85^{th}$ percentiles.

In case of the blade (Fig. 10 (a)), an increase of energy at the blade passing frequency 1P is observed for the waked turbine compared to the freestream turbine. It comes along with a reduction of the blade passing frequency (freestream: $1P \approx 0.15$ Hz,

wake: $1P \approx 0.13$ Hz) due to the reduced wind speed inside the wake and consequently a reduction of rotor speed. The magnitude of the first blade passing frequency of turbine AV5 is predicted by FAST.Farm to be a factor of 3 higher compared to the measurements. In contrast, the excitation of the second blade passing frequency 2P indicated by the peaks between $0.2-0.3$ Hz is higher in the measurements. A possible explanation is the modeling of the wake, which has a Gaussian shape in FAST.Farm. In reality, the wake is more likely to be distorted, leading to smoother transitions to undisturbed winds. Similar observations

are made by Shaler and Jonkman (2020), who compare FAST.Farm with LES. However, more detailed analyses are required, e.g. using LES, to derive a conclusive explanation for the observed characteristics in the frequency response.

The signal at the tower base (Fig. 10 (b)) reveals a strong excitation of the first global mode at around $0.3$ Hz for the waked turbine. This is observed in both the measurements and FAST.Farm, whose prediction of the peak is $15\,\%$ below the measurements. It was found that especially the inclusion of wake-added turbulence in the FAST.Farm simulations has an

intensifying effect on the peak of the first global mode. Higher energy content is also seen in the frequency range $0.35-0.4$ Hz for the waked turbine compared to the freestream turbine. Although FAST.Farm captures an increase in the energy content compared to the freestream turbine, it underestimates the energy level detected in the measurements by $50\,\%$ in this frequency range.

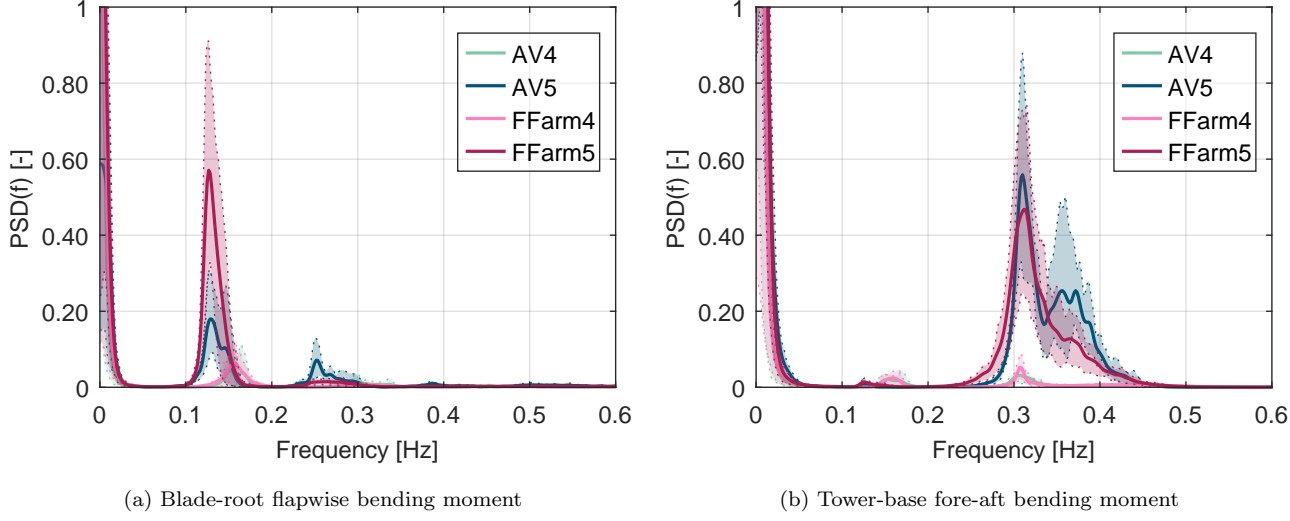

(a) Blade-root flapwise bending moment

(b) Tower-base fore-aft bending moment

**Figure 10.** Comparison of power spectral densities (PSD) between measurements (AV4, AV5) and simulations (FFarm4, FFarm5) for wind speed bin I (wind direction $270.75° - 273.25°$). Thick lines show the mean of the PSDs for all considered events. Shaded area indicates $15^{th}$ and $85^{th}$ percentiles.

## 5 Discussion

The present investigation concentrates on two wind speed bins in below rated conditions. This choice is motivated by the connected high rotor thrust conditions and hence strong wake effects. Moreover, analyses of quantities over wind direction are enabled. Another reason is that the behavior of both turbines AV4 and AV5 is comparable in the measurements, whereas in above rated conditions, differences occur even under ambient inflow for both turbines (see Fig. 3 (c)). For the analysis of structural loads, we focus on sensors and directions that are mainly affected by the change of turbulence characteristics in the
wake, i.e. tower-base FA bending moment and blade-root flapwise bending moment.

A crucial part of this load validation is the generation of adequate wind fields representing the environmental conditions at alpha ventus. Especially, coherence and turbulence scale have an influence on wake meandering magnitude, which in turn affects the loads of the downstream turbine (see also Shaler et al. (2019b) and Wise and Bachynski (2020)). Unstable and neutral atmospheric conditions imply greater turbulent length scales and larger coherent turbulent structures than stable conditions.
This leads to higher wake meandering magnitudes and higher load levels for the downstream turbine. For the loads at the blade-root, adequate capturing of wake meandering is most important, whereas for the loads at the tower-base, both wake meandering and wake-added turbulence must be modeled. We observed that in the simulations, a direct relationship between ambient TI conditions and wake loads exists. Consequently, higher ambient TI values lead to higher loads at the downstream turbine. In the measurements, this relationship holds true but it was also found that low ambient TI conditions can lead to high
wake loads. This shows that there is some uncertainty in modeling the environmental conditions and wake features that should be investigated in future.

In offshore full-scale load validation, there are many potential sources of uncertainty. Starting with the modeling of environmental conditions, we aimed to minimize those uncertainties by making use of findings from previous research, which is available for the site. However, there are limits in the methods used. For example, in the coherence model of the wind
field, there is no directional dependency of coherence considered and coherence is only dependent on the velocity components $u, v, w$. In the considered period of measurements and wind direction sector, alpha ventus operates in the wake of the wind farm "Trianel Borkum I", which is located $\approx 6.5\,\mathrm{km}$ east. The flow structures evolving from this farm-wake are likely to be different from ideal freestream conditions, for which the wind field generation method was originally derived. Overall, to reduce the input uncertainties, we followed a one-to-one simulation approach where the measured environmental conditions are utilized
directly as simulation inputs.

## 6 Conclusions

The simulation tool FAST.Farm was validated for the prediction of power output and structural loads in single wake conditions with respect to measurement data from the offshore wind farm alpha ventus. In addition, a wake-added turbulence model was implemented into FAST.Farm, which is needed to calculate the small-scale turbulence in the wake; this is considered of im-
portance especially for low ambient turbulence intensity conditions and for tower-base loading. It was shown that FAST.Farm predicts the mean power deficit with high accuracy compared to the measurements. Additionally, the probability density func-

tion of generator speed was calculated in strong agreement with the measurements for the freestream and downstream turbine. Fatigue loads were analyzed in terms of damage equivalent loads of the bending moments at the blade root in the flapwise direction and the tower base in the fore-aft direction. Distributions over wind direction show a good match between simulations and measurements with deviations of less than $10\,\%$ for most of the investigated wind directions.

More detailed insights in the aforementioned structural load quantities were provided by power spectral density (PSD) analyses. They show that FAST.Farm calculates trends in the structural response with good agreement to the measurements in the frequency domain. In particular, excitation at the tower base of the waked turbine is reproduced with FAST.Farm, which can be attributed to the wake-added turbulence feature in FAST.Farm added in the course of this study. However, by looking at the PSD at blade root, it is indicated that not all phenomena are captured sufficiently by FAST.Farm, leaving room for further improvements.

It was demonstrated that the proposed one-to-one simulation approach works well for the validation in offshore single wake conditions. It is concluded that calibration of the aeroelastic model with respect to imbalances as well as appropriate transfer of environmental conditions to simulations is important. Here, a differentiation of atmospheric stability helps to refine simulation inputs such as coherence in the wind field, but also indicates that especially stable atmospheric conditions remain challenging to model for capturing the loads of a waked turbine.

*Code and data availability.* The version of FAST.Farm including the wake-added turbulence model, which was used to create the results presented in this paper is different than the official version released by NREL, but is available under Jonkman et al. (2021). NREL has plans to incorporate an updated version of the present implementation in a future release of FAST.Farm

The measurement data used in this study can be accessed via the "Bundesamt für Seeschifffahrt und Hydrographie" (BSH (2021)). A data usage agreement must be signed with the BSH in advance.

*Author contributions.* MK and JJ created with help of VP the conceptualization of this project. MK processed, aggregated, and filtered the measurement data. MK added the wake-added turbulence feature to FAST.Farm, set up and performed all simulations. MK wrote the manuscript with support from JJ. JJ, VP, and PWC provided guidance for the research and reviewed the manuscript.

*Competing interests.* The authors declare that they have no conflict of interest.

*Disclaimer.* The Alliance for Sustainable Energy, LLC (Alliance) is the manager and operator of NREL. NREL is a national laboratory of the U.S. Department of Energy, Office of Energy Efficiency and Renewable Energy. This work was authored in part by the Alliance and supported by the U.S. Department of Energy under Contract No. DE-AC36-08GO28308. Funding was provided by the Wind and Water Power Program. The views expressed in the article do not necessarily represent the views of the U.S. Department of Energy or the U.S.

395 government. The U.S. government retains, and the publisher, by accepting the article for publication, acknowledges that the U.S. government retains a nonexclusive, paid-up, irrevocable, worldwide license to publish or reproduce the published form of this work, or allow others to do so, for U.S. government purposes.

*Acknowledgements.* This study was funded by the German Federal Ministry for Economic Affairs and Energy (BMWi) in the framework of the national joint research project RAVE - OWP Control (FKZ 0324131B) and is part of the research done in the WindForS research cluster.
400     We also would like to thank Senvion and DNV GL for providing the turbine measurements. Additional acknowledgements go to the Federal Maritime and Hydrographic Agency (BSH) for providing access to the FINO1 data base and UL DEWI for providing wind measurement corrections of the FINO1 met mast.

## Appendix A

### A1    Influence of the wake-added turbulence feature in FAST.Farm

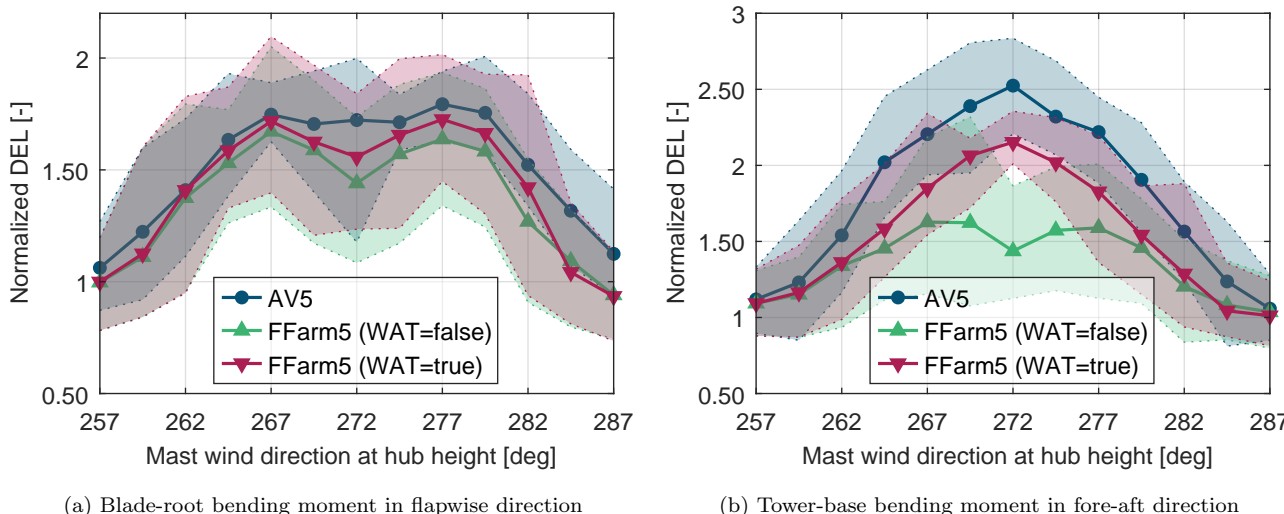

(a) Blade-root bending moment in flapwise direction      (b) Tower-base bending moment in fore-aft direction

**Figure A1.** Comparison of FAST.Farm simulations with activated (WAT=true) and deactivated (WAT=false) wake-added turbulence with respect to the measurements (AV5) for wind speeds $7.5-8.5\,\mathrm{m\,s^{-1}}$. Statistics are shown as mean values per wind direction bin. Shaded area indicates $15^{th}$ and $85^{th}$ percentiles.

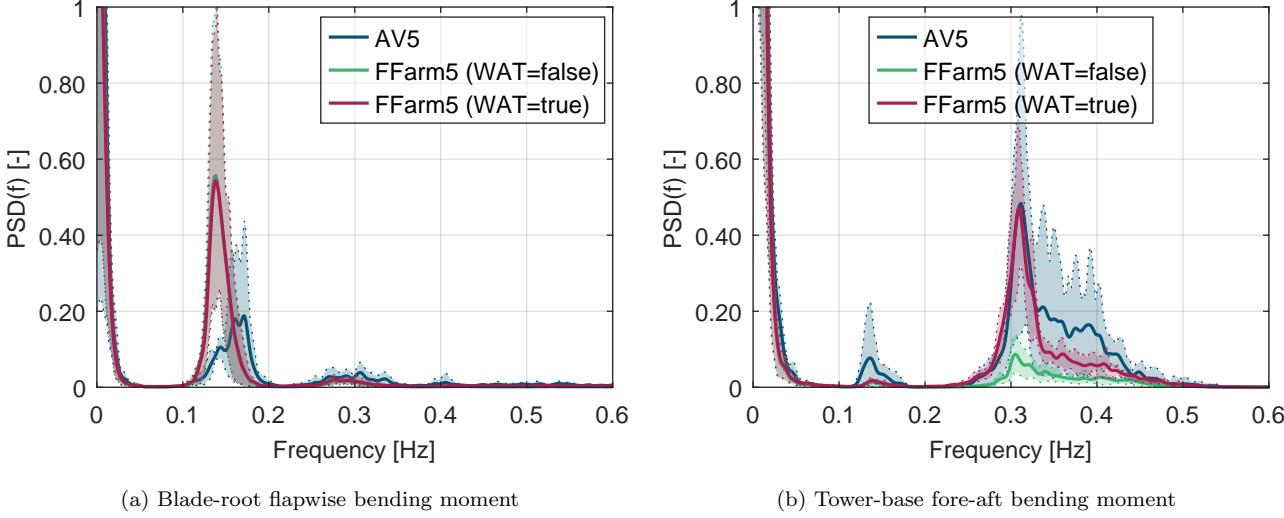

(a) Blade-root flapwise bending moment         (b) Tower-base fore-aft bending moment

**Figure A2.** Comparison of power spectral densities (PSD) between measurements (AV5) and FAST.Farm simulations with activated (WAT=true) and deactivated (WAT=false) wake-added turbulence for wind speeds $7.5 - 8.5\,\mathrm{m\,s^{-1}}$ (wind direction $270.75° - 273.25°$). Thick lines show the mean of the PSDs for all considered events. Shaded area indicates $15^{th}$ and $85^{th}$ percentiles.

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
