# Peer review of "FAST.Farm load validation for single wake situations at alpha ventus"

_Wind Energy Science, 2021_

## Author Comment (AC1)

**Reviewer 1 (Anonymous)**

We thank the reviewer for the effort and the valuable comments that improved the quality of the paper. We have addressed all the comments from the reviewer and modified the manuscript accordingly. The responses to the review are marked in blue.

**Comment nr 1:**

The use of TurbSim to model the input wind field based on measurements for 1-to-1 simulations is presented in section 2.4, and discussed in section 5. While high-quality modeling and synchronization with measurements is ensured for mean wind speed and power spectral density, coherence has received less attention, with a number of caveats :

- TurbSim uses Taylor's frozen wake assumption which is not valid for this turbine configuration (the input wind field on AV5 is not simply the field on AV4 with a delay, see for instance Vigueras-Rodriguez et al., 2012 for a farm-wide coherence model).

  We agree that Taylor's frozen turbulence assumption is not completely realistic because the evolution of turbulent eddies over time is neglected. However, the impact of wind evolution is considered to be small compared to the impact of the wake on the downstream turbine. This is confirmed by Shaler et al. (2019) where different inflow generation techniques are investigated; they compare results from a FAST.Farm simulation with a LES-generated wind field (effectively applying Taylor's frozen turbulence) to results from a fully-coupled LES with the same LES precursor. They conclude that Taylor's frozen turbulence assumption is reasonable by analyzing the wake center displacements at various downstream distances. We incorporated this clarification in the manuscript in Section 2.4.

- Coherence is modeled using a statistical model, not the actual 10-min observations. The wind field is not a reconstruction around the observations that would be consistent with the 1-to-1 simulation approach.

  It is true that the values for the coherence model are based on long-term investigations by Nybø et al. (2020) which are made for the FINO1 site. We consider the usage of these statistical values as a good approximation for the site-specific conditions. Eventually, the paper presents statistical analyses and not direct comparison of time series, where we agree that the coherence of a specific event would be considered of major importance.

- The way 2.4 is written lets the reader believe that u,v,w components are correlated. In reality, only the TIs are correlated, not the realization. The coupling between u and w coherence (see for instance Cheynet et al, 2018 for FINO1 data) is to my knowledge not implemented in TurbSim.

*We agree with the reviewer and added a clarification regarding the coherence model in Section 2.4.*

It might be argued that the impact of those caveats is low as the focus is not on time-domain nor cross-spectral comparison, and when compared to the overall scatter. Still this should be made clear.

*We agree and have made the corresponding clarifications in the manuscript (see points above).*

**Comment nr 2:**

In section 3.2, given the scatter on Fig. 3, the only valid conclusion may be that the turbulence intensity cannot be modeled using only mean speed, direction and stability class as parameters. Suggestions would be welcome.

*Figures 3 and 4 are presented in order to provide the reader a visualization of the distribution of important environmental parameters as well as their variation. They are used as direct inputs for the generation of wind fields. It is not aimed to derive relationships for the turbulence intensity with respect to for instance atmospheric stability or wind direction.*

**Comment nr 3:**

In section 4.4, the relationship between a skewed wake and an increase in 2P excitation combined with a decrease in 1P is not trivial. The given reference does not appear to provide more information. A better explanation would be welcome.

*We believe that more detailed analyses, e.g. using LES, are necessary to give a conclusive explanation on the reported characteristics in the frequency response. For the moment, we could not come up with a more conclusive explanation and can only provide the given possible explanation. We updated the manuscript accordingly, stating that the given explanation is tentative and that further analyses are required.*

**References:**

Shaler, K., Jonkman, J., Doubrawa, P., & Hamilton, N. (2019). *FAST.Farm Response to Varying Wind Inflow Techniques: Preprint*. 37th Wind Energy Symposium, San Diego, CA. https://arc.aiaa.org/doi/pdf/10.2514/6.2019-2086

Nybø, A., Nielsen, F. G., Reuder, J., Churchfield, M. J., & Godvik, M. (2020). Evaluation of different wind fields for the investigation of the dynamic response of offshore wind turbines. *Wind Energy*, *23*(9), 1810–1830. https://doi.org/10.1002/we.2518

**Reviewer 2 (Adam Wise)**

We thank Adam Wise for the effort and the valuable comments that improved the quality of the paper. We have addressed all the comments from the reviewer and modified the manuscript accordingly. The responses to the review are marked in blue.

**General Comments**

This paper aims to validate the tool FAST.Farm for single wake situations at the Alpha Ventus Wind Farm. The novelty in this work is that a new wake-added turbulence model is added to the tool FAST.Farm, which improves agreement especially in lower wind speeds when the wake-added contribution of turbulence is most significant. This paper analyzes measurements from the FINO1 met mast just upstream of the farm to determine the environmental conditions for the simulations. The data is filtered and categorized by stability and wind speed, then statistics are fed into TurbSim to conduct "one-to-one" simulations. The paper is mostly well-written; however, the manuscript needs minor proofreading/editing to correct small grammatical errors throughout. This work improves FAST.Farm's usability, as validation with utility-scale measurements, such as tower and blade loads in this case, is very important. Specific scientific questions and a number of technical suggestions are listed below.

**Specific Suggestions and Questions**

1.  It would help to provide more detail on the implementation of wake-added turbulence so that the reader does not need to open another reference or the IEC standard. The implementation of wake-added turbulence is one of the more important aspects of this paper so more detail should be provided in Section 2.1.1.

    We added more information on the actual implementation of wake-added turbulence into the FAST.Farm code in Section 2.1.1.

2.  Line 82: Wake-added turbulence is generated using the Mann Model, but the ambient wind is generated using TurbSim. Please comment on why this is appropriate. Why is wake-added turbulence not just added with TurbSim? There are fundamental differences in the coherent structures between the two methods for generating synthetic turbulence (perhaps cite Bachynski and Eliassen 2018 as well as the already cited Nybø 2020).

    The wake-added turbulence is considered independent from the ambient turbulence because it includes contributions from mechanically generated shear, caused by the wake deficit, as well as the breakdown of mainly the tip and root vortices. The homogenous and isotropic wake-added turbulence is scaled with the radially dependent factor $k_{mt}$, thus second order statistics are violated.
    We agree that there are fundamental differences in the coherent structures between the Mann and Veers method. However, given the assumptions for the wake-added

turbulence, these differences are considered of minor importance and the usage of a wake-added turbulence field generated with TurbSim should give comparable results.

The choice of using the Mann model for generating the wake-added turbulence box is based on practical reasons concerning the implementation in FAST.Farm. When using the Mann model, the turbulence box is saved without incorporating a mean wind profile. The mean wind profile can be added inside the InflowWind module of FAST.Farm. However, the wake-added turbulence model does not require a mean wind profile, but it needs the definition of a velocity that is used to propagate the wake-added turbulence wind field. This can be defined quite easily within InflowWind, which is used in a second instance for the wake-added turbulence domain in addition to the InflowWind instance already used for ambient turbulence. The usage of a wake-added turbulence domain generated with TurbSim would require additional information exchange with the wake-added turbulence InflowWind instance, which is avoided in the current implementation.

We added more information on the implementation of the wake-added turbulence in the manuscript in Section 2.1.1.

3. Section 2.1.1: Why are the empirical coefficients the authors calibrated so different than the recommended values by the IEC? Is this site or turbine specific? Also, please provide example calculations for Eq. 1 so that the reader has an idea of how much the velocity components might scale for the given environmental conditions.

   The reason for the discrepancy between the IEC values and the re-calibrated values is not clear. Different implementations of the wake-added turbulence model might have an influence, but the implementation in Madsen et al. (2010) that is used as reference in the IEC is not available to us. We added supplementary plots that provide the requested exemplary calculations of the wake-added turbulence model in Fig. 1. We also compared the resulting turbulence levels from the wake-added turbulence in Fig. 1 with levels given in the literature (Madsen et al. (2010), Keck et al. (2014)). We found that the turbulence levels are similar and conclude that the implementation in FAST.Farm is reasonable.

4. Additionally regarding Section 2.1.1, the figures in Appendix A would be more helpful if they are moved to the body of the paper, i.e. to Section 2.1.1. Additionally, please add PSDs for FAST.Farm to show the difference in the energy content with and without wake-added turbulence.

   We added the requested PSDs for FAST.Farm in Fig. A2. We decided to keep the Figures in the appendix as supplementary material because necessary definitions for the interpretation of these figures (e.g. DEL) are missing in the manuscript at the desired position in Section 2.1.1.

5. Figure 2: please comment on why there are discrepancies for OpenFAST compared to AV4 for wind speeds above 16 m/s. Alternatively, the data from wind speeds above 16 m/s

could be removed. If the data are included, the paper should discuss why OpenFAST is underperforming in this regime.

We had a look at the data again, but could not find a conclusive explanation for the observed discrepancies. Eventually, we decided to remove the data above 16 m/s and updated the plots accordingly.

6. Section 2.6: Please provide information regarding the grid resolution used for the setup of the simulations. Details are needed for the dx, dy, dz used for the turbsim wind fields, the Mann wind fields (used for the wake-added turbulence), the low-res domain in FAST.Farm and for the high-res domain around each turbine. If the simulations are run for just 10 minutes because 10-minute statistics are used, the paper should state that each realization from TurbSim is 10 minutes long. Is there a spin-up period included, since it takes some time for the wake to advect from AV4 to AV5? These are critical parameters that add to the credibility of the model. The paper should describe the research done such that the modeling details are clear and reproducible by future researchers.

We added the necessary information on the setup of the FAST.Farm simulations and wind fields in Section 2.6.

**Minor comments**

- Lines 5 and 6: Please quantify the agreement between FAST.Farm and the measurements in the abstract. It is added to the manuscript.
- Line 40 (and so on): Fino 1 should be written as FINO1. It is corrected in the manuscript.
- Line 93 (and so on): Alpha Ventus should be capitalized. Apparently, the name alpha ventus is uncapitalized by definition, so we have not changed it.
- Line 94: Remove the word "form". Removed.
- Line 96: "Research" should not be capitalized. It is uncapitalized now.
- Table 1 should mention the Pena reference for how sigma_v and sigma_w are determined. We added the reference here.
- Line 133: "Sea state is measured in terms of significant wave height and peak wave period." How were the wave height and peak wave period measured at FINO1? Please state this in the manuscript. The measurement device is a directional waverider buoy; this information is added to the manuscript.
- Line 183: Are six random and uncorrelated sea states also used for the simulations? It's unclear how the wave loading is represented for both the OpenFAST and the FAST.Farm simulations. Please clarify. This is clarified by adding additional information on the sea state modelling.
- Line 204: Could the freestream sector of 240-257 deg be shaded in a separate color for Figure 1. We prefer not to mark the sector in this figure to maintain readability and set the focus on the wake situation.
- Line 259: approximately should be spelled out and factor 2 should be factor 2.0. It is corrected in the manuscript.

- Line 263: again should be factor 2.0. It is corrected in the manuscript.
- Line 286: The higher energy content from 0.35-0.4 Hz is 3P, correct? Any ideas on why FAST.Farm is underestimating this excitation? It is true that the frequency range 0.35-0.4 Hz corresponds to the 3P excitation. For now, we do not have a conclusive explanation for the underestimation in FAST.Farm, given the fact that the 1P excitation in the rotating frame of reference is overestimated by FAST.Farm.
- Line 295: In addition to Shaler et al. 2019, Wise et al. 2020 should also be cited as it discusses the effect of coherence on wake meandering for the DWM in FAST.Farm. We have added the suggested reference as it gives additional insights into the influence of coherence on the wake meandering.
- Line 299: larger coherent structures, not necessarily more. Agreed and changed accordingly.
- Line 311: km should not be italicized. It is updated in the manuscript.
- Line 312: remove the word "order". It is corrected.
- Line 319: "free- and downstream" should be freestream and downstream. It is updated in the manuscript.
- Line 326: Please add a sentence that succinctly describes the wake-added turbulence method used in this paper. We added a sentence in the conclusions.
- Careful proofreading to correct minor grammatical errors is needed throughout the manuscript. The paper has now been edited by the NREL communications team.

**References**

Madsen, H. Aa., Larsen, G. C., Larsen, T. J., Troldborg, N., & Mikkelsen, R. (2010). Calibration and Validation of the Dynamic Wake Meandering Model for Implementation in an Aeroelastic Code. *Journal of Solar Energy Engineering*, *132*(4), 041014. https://doi.org/10.1115/1.4002555

Keck, R.-E., de Maré, M., Churchfield, M. J., Lee, S., Larsen, G., & Aagaard Madsen, H. (2014). On atmospheric stability in the dynamic wake meandering model: On atmospheric stability in the dynamic wake meandering model. *Wind Energy*, *17*(11), 1689–1710. https://doi.org/10.1002/we.1662